# Joint Resource Allocation in TWDM-PON-Enabled Cell-Free mMIMO System

**Tianyu Xue [1], Kamran Ali Memon [2,3] and Chunguo Li [4,\*]**

[1] School of Materials and Metallurgy, Wuhan University of Science and Technology, Wuhan 430081, China; xuety@wust.edu.cn
[2] Center for Communication Systems and Sensing, King Fahd University of Petroleum and Minerals, Dhahran 31261, Saudi Arabia; alikamran77@quest.edu.pk
[3] Department of Telecommunication Enigneering, QUEST University, Sindh 67450, Pakistan
[4] School of information science and engineering, Southeast University, Nanjing 210000, China
\* Correspondence: chunguoli@seu.edu.cn

**Abstract:** Cell-free massive multiple input multiple outputs (CF-mMIMO) is considered a promising technology for sixth-generation (6G) telecommunication systems. In the CF-mMIMO system, an extensive array of distributed small base stations (BSs) is deployed across the network, which enables us to facilitate seamless collaboration among BSs. To achieve this goal, the baseband signal from these BSs needs to be transmitted to a central server via fronthaul networks. Due to the large number of BSs, the data that needs to be transmitted is usually huge, which brings severe requirements on fronthaul networks. Time and wavelength division multiplexed passive optical networks (TWDM-PON) can be a potential solution for CF-mMIMO fronthaul due to their large capacity and high flexibility. However, how to efficiently allocate both optical and wireless resources in a TWDM-PON-enabled CF-mMIMO system is still a problem to be addressed. This paper proposes a joint scheduling method of wavelength, antenna, radio unit (RU), and radio resource block (RB) resources in the TWDM-PON-enabled CF-mMIMO system. Furthermore, an integer linear programming (ILP) model for joint resource allocation is proposed to minimize the fronthaul resource occupancy, thereby increasing network scalability. Considering the complexity of the ILP model, two heuristic algorithms are also presented to solve this model. We compare the ILP with heuristic algorithms under different scenarios. Simulation results show that the proposed algorithm can reduce the fronthaul resource occupancy to improve the network scalability of the CF-mMIMO system.

**Keywords:** 6G; CF-mMIMO; TWDM-PON; joint resource allocation

## 1. Introduction

6G is considered the next generation of communication technology beyond 5G. It is expected to introduce revolutionary advancements in wireless communication to meet the increasing demands for higher data rates, lower latency, enhanced spectral efficiency, and support for a massive number of connected devices [1]. Cell-free massive multiple input multiple output (CF-mMIMO) is considered a promising technology to support 6G due to its potential to significantly improve network performance. As shown in Figure 1, a CF-mMIMO system consists of a central processing unit (CPU), which is connected to a large number of distributed radio units (RUs) via the fronthaul [2]. In CF-mMIMO, the RUs serve a group of user equipment (UEs) over the same time-frequency resource, and the number of RUs should exceed that of UEs. Compared with cellular mMIMO [3–5], CF-mMIMO can improve the quality of services for cell-edge users and reduce intercell interference of cellular systems [6]. In early research on CF-mMIMO, each UE is assumed to be served by all RUs [7,8], which is not realistic because it leads to a significant waste of resources and high computational complexity. For this, [9] proposed a conception of user-centric CF-mMIMO, where each UE is served by a specific subset of Rus, providing

the best channel conditions. The user-centric CF-mMIMO improves operational efficiency, especially when serving massive UEs, and it has been regarded as a practical way to deploy cell-free communication.

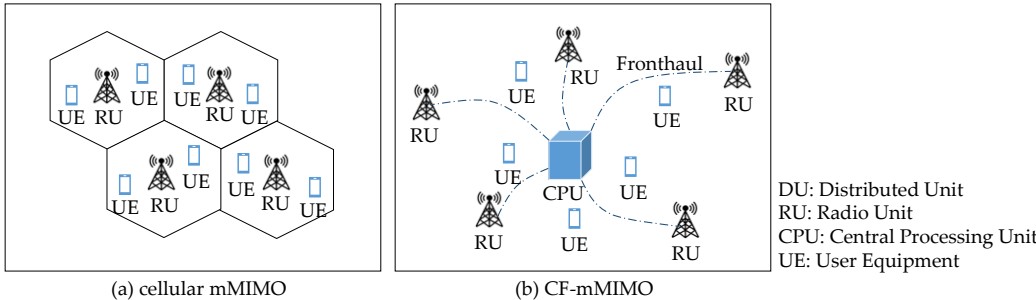

**Figure 1.** (**a**) Architecture of cellular mMIMO systems. (**b**) Architecture of CF-mMIMO systems.

On the other hand, the fronthaul plays a crucial role in a CF-mMIMO system as it facilitates the exchange of data, including the control information between the distributed RUs and the central controller [10]. The fronthaul requires a high capacity, as it does not only carry the signals received from multiple RUs to the central processor for joint processing but also delivers the processed data back to the RUs for transmission to the UEs. As a primary technology of the next generation PON stage 2 (NG-PON2), time and wavelength division multiplexed passive optical network (TWDM-PON) combines both time division multiplexing (TDM) and wavelength division multiplexing (WDM) technologies, enabling multiple wavelengths to be transmitted over the same fiber optic cable [11]. A TWDM-PON has multiple wavelength channels, thus supporting a higher capacity of the system [12]. Therefore, TWDM-PON is a potential solution for the fronthaul in the CF-mMIMO system, and a TWDM-PON-based CF-mMIMO can accommodate the massive data flows generated by the distributed RUs, thus meeting the large demand for the system capacity in CF-mMIMO. However, the growing number of wireless devices (i.e., RUs and UEs) brings a great challenge of scalability to the CF-mMIMO system, which makes it difficult for CF-mMIMO to effectively deliver higher data rates and improved user experiences [13].

It is necessary to adopt an efficient resource allocation scheme to enhance the system scalability in CF-mMIMO. In terms of this, several works on the resource allocation schemes in CF-mMIMO have been proposed. Jung et al. [14] proposed a RU selection scheme in CF-mMIMO using a linear assignment algorithm. Guenach et al. [15] proposed a RU selection and power allocation algorithm, and Mazhari Saray et al. [16] proposed a deep learning method to solve power allocation in the uplink mode of the CF-mMIMO system. However, all these works focused on the single radio resource allocation in CF-mMIMO, while the joint optimization of both optical and radio resources in CF-mMIMO is rarely studied. Additionally, there have been several works on joint resource optimization in the mMIMO system. Zhang et al. [17] proposed three heuristic algorithms in mMIMO to optimize the fronthaul bandwidth, the utilization of radio resource blocks (RBs), and both of them. Wang et al. [18] proposed a deep reinforcement learning-based joint resource allocation policy for mMIMO-enabled beaming in a fronthaul network. However, the RU assigned to UEs is selected among all RUs in mMIMO, while in CF-mMIMO, RUs can only be selected from a specific RU set. In addition, no scheme on wavelength allocation is mentioned in [17,18]. Thus, all these algorithms cannot be directly applied in CF-mMIMO.

In this paper, we propose a joint resource allocation scheme in a TWDM-PON-enabled CF-mMIMO system to increase network scalability. First, we formulate a joint optimization problem (involving the allocation of wavelength, antenna, RU, and radio RB), and we solve it with an ILP model. In order to reduce the executing time for large-scale networks, two heuristic algorithms are proposed, i.e., the Random Resource Allocation (RRA) algorithm and the Fronthaul Occupancy Minimized Resource Allocation (FOM-RA) algorithm. Simulation results are presented to compare the performance of ILP and heuristic algo-

rithms under different simulation scenarios. Simulation results show that FOM-RA can effectively optimize the fronthaul bandwidth and wavelength usage compared with RRA. Additionally, compared with ILP, FOM-RA can reduce the execution time.

The rest of the paper is arranged as follows: Section 2 details the architecture of the TWDM-PON-enabled CF-mMIMO system and the joint resource scheduling method in this system. Section 3 gives an ILP formulation for the joint resource allocation. In Section 4, two heuristic algorithms are proposed to solve the ILP model. In Section 5, simulation results are presented to compare the performance of ILP and the heuristic algorithms under different scenarios. Section 6 concludes the paper.

## 2. TWDM-PON-Enabled CF-mMIMO System

In this section, we introduce the architecture of the TWDM-PON-enabled CF-mMIMO system, and then a joint resource scheduling method with specific constraints is presented based on this system.

The architecture of TWDM-PON-enabled CF-mMIMO is shown in Figure 2. The system comprises a distributed unit (DU) pool, where the enhanced common public radio interface (eCPRI) signals are processed. The DU pool is connected to an optical line terminal (OLT), which accounts for receiving upstream data from the DU pool and sending downstream data to optical network units (ONUs). We noted that a controller exists between the DU pool and the OLT. It obtains the status information of optical and wireless resources from the two devices through the device interface, which includes the remaining bandwidth of the fronthaul, the load of each RU, and the number of requested UEs [19]. After performing the cooperative allocation of both resources with its joint resource allocation module, the controller sends the wireless and optical scheduling messages back to the DU pool and the OLT, respectively. The OLT is equipped with a wavelength division multiplexing multiplexer (WDM MUX) and a wavelength division multiplexing demultiplexer (WDM DEMUX) [20]. WDM MUX combines multiple optical signals within different wavelengths onto a single optical fiber, which are represented as $\{\lambda_1, \lambda_1'\}, \{\lambda_2, \lambda_2'\} \ldots \{\lambda_n, \lambda_n'\}$, allowing the coexistence of multiple wavelengths in the optical channel, and WDM DEMUX accounts for separating the combined multiple wavelengths from the optical fiber into individual optical channels. This enables the upstream and downstream data transmission with multiple wavelengths between the OLT and the ONUs [21]. ONUs are equipped with a tunable transmitter and transceiver that can be tuned to any wavelength in both downstream and upstream directions [22]. The downstream data is received by each ONU within a particular wavelength because an ONU can be assigned one or multiple wavelengths in a TWDM-PON-based fronthaul, and one wavelength can be shared with multiple ONUs. Each ONU is connected to a corresponding RU, and each RU is equipped with one or multiple antennas. Additionally, since the TWDM-PON-enabled CF-mMIMO system is a user-centric CF-mMIMO system, it defines a serving RU cluster for each UE, and each UE is served by RUs in this cluster, providing the best channel conditions [23]. As shown in Figure 2, the red-colored and blue-colored dash lines illustrate the RU clusters that serve different UEs.

The controller plays a critical role in the control plane of the TWDM-PON-enabled CF-mMIMO system because it is responsible for the joint allocation of optical and wireless resources, including wavelength, antenna, RU and radio RB. Figure 3 shows an example of the joint resource scheduling method in the TWDM-PON-enabled CF-mMIMO system, which is divided into three steps. In the first step, the scheduling process involves assigning RUs along with their corresponding antennas to each UE. Consider a scenario where the RU cluster assigned to UE 1–3 is labeled as RU 1–3. The distribution of RUs and the corresponding antennas allocated to the various UEs is illustrated in Figure 3. Notably, the number of selected antennas is subject to the UEs' specific requests. In the second step, the scheduling process involves assigning RBs within their antennas to each UE. Different UEs cannot reuse the RBs of the same antenna. Additionally, taking into the interference between UEs, it is important to avoid allocating the same RBs to UEs that are close to each

other. Assume that the number of requested RBs is 3, and the distance between UE 1 and 2 is relatively close, and UE 3 is far away from each of them. Once RBs 1–3 have been assigned to UE 1, only RBs 4–6 remain available for allocation to UE 2. Additionally, the RB allocation for UE 3 remains unaffected by the allocations for either UE 1 or UE 2. In the third step, the scheduling process involves assigning the wavelength of the TWDM-PON to each RU. Each RU is limited to utilizing a single wavelength, although a single wavelength can be allocated to multiple RUs. It is noted that the sum of the occupied bandwidth of a wavelength cannot exceed the maximal capacity of the wavelength. As shown in Figure 3, wavelength $\lambda_1$ is allocated to RU 1 and RU 2 because the capacity of $\lambda_1$ could satisfy the total bandwidth of them. However, the remaining capacity is not sufficient for the bandwidth requirement in RU 3, so $\lambda_2$ is allocated to RU 3. In this procedure, the first and second steps involve scheduling wireless resources; thus, the generated scheduling messages are transmitted to the DU. The third step pertains to scheduling optical resources, and the scheduling messages are transmitted to the OLT. After this, the DU and the OLT manage and schedule their RUs and ONUs according to the messages, respectively.

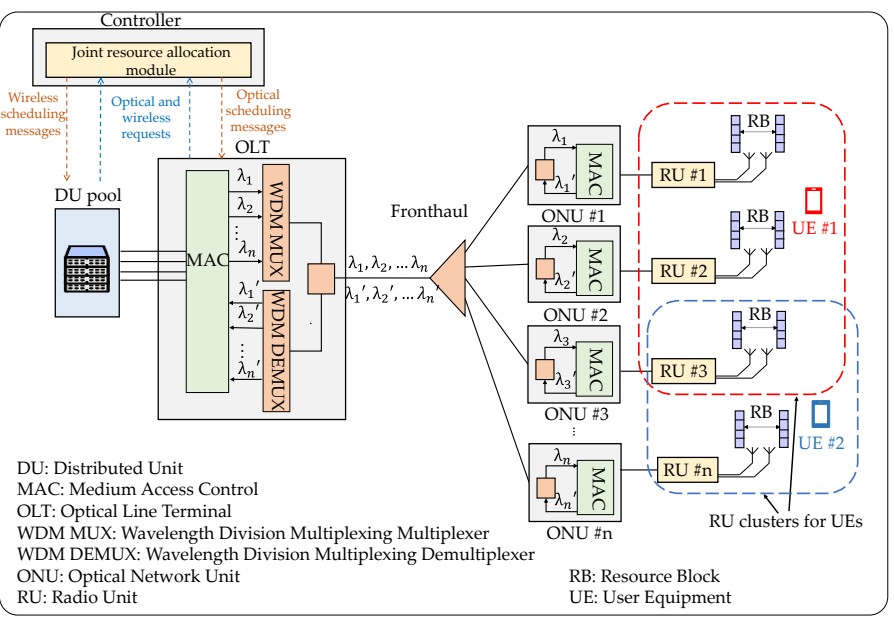

**Figure 2.** TWDM-PON-enabled CF-mMIMO system.

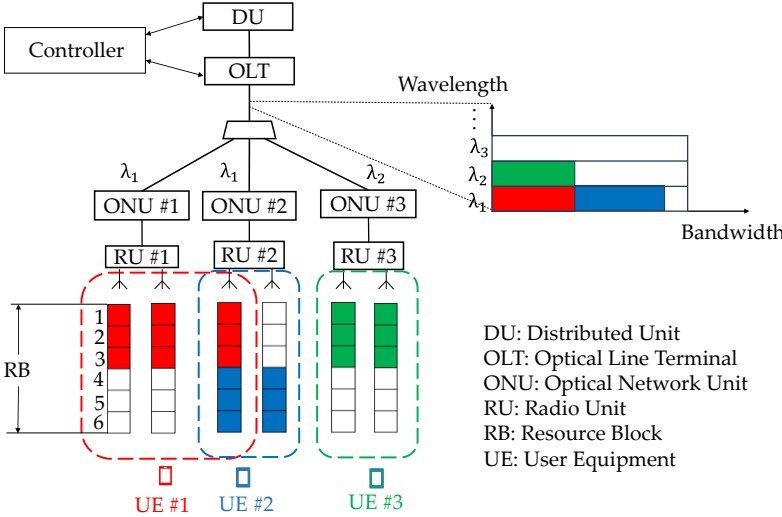

**Figure 3.** Joint resource allocation in TWDM-PON-enabled CF-mMIMO system.

## 3. ILP Formulation

Based on the joint resource scheduling method, we further propose an ILP formulation for joint resource allocation in the TWDM-PON-enabled CF-mMIMO system, which minimizes fronthaul resource occupancy to increase network scalability while satisfying specific constraints. First, we define the notations of parameters and variables in the ILP, and then the constraints and the objective function are detailed.

### 3.1. Parameters

The parameters in ILP are listed as follows:

- $N$, the set of RUs, $\{n|n \in N\}$.
- $K$, the set of antennas in a RU, $\{k \mid k \in K\}$.
- $M$, the set of UEs, $\{m \mid m \in M\}$.
- $W$, the set of RBs, $\{w \mid w \in W\}$.
- $W_m$, the number of RBs required by UE $m$.
- $L$, the set of wavelengths in a TWDM-PON, $\{l \mid l \in L\}$.
- $K_m$, the number of antennas required by UE $m$.
- $P_m$, the power of an RB.
- $P_{max}$, the power threshold of an RU.
- $R_m$, the transmission rate of an RB.
- $A_m$, the set of RUs that serve UE $m$.
- $R_o$, the optical transmission rate in a single wavelength.
- $r_m$, the bandwidth requirement of UE $m$.
- $\Delta_{ma,mb}$, binary, which represents the positional relationship between two UEs, $\{ma, mb \mid ma, mb \in M\}$. If $\Delta_{ma,mb}$ is equal to 1, the distance between UE $ma$ and $mb$ is close.

### 3.2. Variables

The variables in ILP are as follows:

- $\delta_{n,k}^m$, binary, which represents the situation in which the RUs and antennas are allocated to the UEs, where $m \in M$, $n \in N$, and $k \in K$. If $\delta_{n,k}^m$ is equal to 1, RU $n$ and RB $w$ are allocated to UE $m$; otherwise, $\delta_{n,k}^m$ is equal to 0.
- $\delta_w^m$, binary, which represents the situation in which the RBs are allocated to the UEs, where $m \in M$, and $w \in W$. If $\delta_w^m$ is equal to 1, RB $w$ is allocated to UE $m$; otherwise, $\delta_{n,k}^m$ is equal to 0.
- $\delta_l^n$, binary, which represents the situation in which the wavelengths are allocated to the RUs, where $n \in N$, and $l \in L$. If $\delta_l^n$ is equal to 1, wavelength $l$ is allocated to RU $n$; otherwise, $\delta_{n,k}^m$ is equal to 0.
- $\delta_n^m$, binary, which represents the situation in which the antennas are allocated to UE $m$, where $m \in M$, and $n \in N$. If $\delta_n^m$ is equal to 1, RU $n$ is allocated to UE $m$; otherwise, $\delta_{n,k}^m$ is equal to 0.
- $\delta_{n,l}^m$, binary, which represents the situation in which the RUs and wavelength are allocated to UE $m$, where $m \in M$, $n \in N$, and $l \in L$. If $\delta_{n,l}^m$ is equal to 1, RU $n$ and wavelength $l$ are allocated to UE $m$; otherwise, $\delta_{n,k}^m$ is equal to 0.
- $\delta_l^m$, binary, which represents the situation in which the wavelengths are allocated to UE $m$, where $m \in M$, and $l \in L$. If $\delta_l^m$ is equal to 1, wavelength $l$ are allocated to UE $m$; otherwise, $\delta_{n,k}^m$ is equal to 0.
- $\delta_{n,k,w}^m$, binary, which represents the situation in which the RUs, antennas and RBs are allocated to UE $m$, where $m \in M$, $n \in N$, $k \in K$, and $w \in W$. If $\delta_{n,k,w}^m$ is equal to 1, RU $n$ are allocated to UE $m$, RB $w$ and RB $w$; otherwise, $\delta_{n,k}^m$ is equal to 0.

- $\delta_l$, binary, which represents the occupancy of wavelength in TWDM-PON, where $l \in L$. If $\delta_l^m$ is equal to 1, wavelength $l$ in TWDM-PON is occupied; otherwise, $\delta_{n,k}^m$ is equal to 0.

*3.3. Constraints*

The constraints in ILP are as follows:

$$\delta_{n,k}^m = 0, \quad \forall n \notin A_m, \forall m \in M, \forall n \in N. \tag{1}$$

Equation (1) guarantees that UE $m$ can only be served by RUs in $A_m$.

$$\sum_{n \in N, k \in K} \delta_{n,k}^m = K_m, \quad \forall m \in M. \tag{2}$$

Equation (2) guarantees that the number of antennas allocated by UE $m$ is equal to the number of antennas requested.

$$\sum_{w \in W} \delta_w^m = W_m, \quad \forall m \in M. \tag{3}$$

Equation (3) guarantees that the sum of RBs in the antenna occupied by UE $m$ should be equal to the total number of RBs requested by UE $m$.

$$\delta_w^{ma} + \delta_w^{mb} \leq 1, \Delta_{ma,mb} = 1, \forall ma, mb \in M, \forall w \in W. \tag{4}$$

Equation (4) guarantees that UEs with a close distance cannot choose the same RB. This ensures that UEs at a close distance are not interfered with by each other.

$$\delta_w^{ma} + \delta_w^{mb} + \delta_{n,k}^{ma} + \delta_{n,k}^{mb} \leq 3, \quad \forall ma, mb \in M, \forall w \in W, \forall n \in N, \forall k \in K. \tag{5}$$

Equation (5) guarantees that different UEs do not reuse the RBs of the same antenna.

$$\delta_{n,k,w}^m \leq \delta_{n,k}^m, \quad \forall m \in M, \forall n \in N, \forall k \in K, \forall w \in W. \tag{6}$$

$$\delta_{n,k,w}^m \leq \delta_w^m, \quad \forall m \in M, \forall n \in N, \forall k \in K, \forall w \in W. \tag{7}$$

$$\delta_{n,k,w}^m \geq \delta_{n,k}^m + \delta_w^m - 1.5, \forall m \in M, \forall n \in N, \forall k \in K, \forall w \in W. \tag{8}$$

$$\sum_{m \in M} \sum_{k \in K, w \in W} (\delta_{n,k,w}^m \times P_m) \leq P_{max}, \quad \forall n \in N. \tag{9}$$

Equations (6)–(8) calculate the situation in which the RUs, antennas and RBs are allocated to UE. Equation (9) guarantees that the total power of the antenna serving the UEs in an RU does not exceed the maximum power limit.

$$\sum_{l \in L} \delta_n^l = 1, \quad \forall n \in N. \tag{10}$$

$$\delta_l \geq \delta_l^n, \quad \forall n \in N, \forall l \in L. \tag{11}$$

$$\delta_l \leq \sum_{n \in N} \delta_l^n, \quad \forall l \in L. \tag{12}$$

Equations (10)–(12) guarantee that each RU can only be allocated to one wavelength in the TWDM-PON.

$$\delta_n^m \geq \delta_{n,k}^m, \quad \forall m \in M, \forall n \in N, \forall k \in K. \tag{13}$$

$$\delta_n^m \leq \sum_{k \in K} \delta_{n,k}^m, \quad m \in M, \forall n \in N. \tag{14}$$

$$\delta_{n,l}^m \leq \delta_n^m, \quad \forall m \in M, \forall n \in N, \forall l \in L. \tag{15}$$

$$\delta_{n,l}^m \leq \delta_l^n, \quad \forall m \in M, \forall n \in N, \forall l \in L. \tag{16}$$

$$\delta_{n,l}^m \geq \delta_n^m + \delta_l^n - 1.5, \quad \forall m \in M, \forall n \in N, \forall l \in L. \tag{17}$$

$$\delta_l^m \geq \delta_{n,l}^m, \quad \forall m \in M, \forall n \in N, \forall l \in L. \tag{18}$$

$$\delta_l^m \leq \sum_{n \in N} \delta_{n,l}^m, \quad \forall m \in M, \forall l \in L. \tag{19}$$

$$r_m = W_m \times R_m, \quad \forall m \in M. \tag{20}$$

$$\sum_{m \in M} \delta_l^m \times r_m \leq R_o, \quad \forall l \in L. \tag{21}$$

Equations (13)–(19) calculate the situation in which the wavelengths are allocated to UE. Equation (20) calculates the bandwidth requested by each UE. Equation (21) guarantees that the sum of occupied bandwidth in a wavelength does not exceed the maximal capacity of a wavelength.

*3.4. Objective Function*

Minimize:

$$\alpha \sum_{m \in M, l \in L} \delta_l^m \times r_m + \beta \sum_{l \in L} \delta_l. \tag{22}$$

The objective function of the ILP is presented in Equation (21), which aims at minimizing the fronthaul resource occupancy. The objective comprises two parts: the first part (i.e., $\alpha \sum_{m \in M, l \in L} \delta_l^m \times r_m$) is to minimize the total fronthaul bandwidth of each wavelength, and the second part (i.e., $\beta \sum_{l \in L} \delta_l$) is to minimize the number of occupied wavelengths. The two parts are linearly summed up by multiplying the weighting factors $\alpha$ and $\beta$ to integrate different dimensions of objective function.

## 4. Heuristic Algorithm

With the expansion of the network scale, solving the ILP problem becomes excessively time-consuming. In this section, two heuristic algorithms are proposed to reduce the execution time to improve efficiency.

The first algorithm we proposed is referred to as Random Resource Allocation Algorithm (RRA), which is based on a random allocation policy. As one of the simplest heuristic algorithms, the random allocation policy can be directly applied in the CF-mMIMO system to solve our joint resource allocation problem. The pseudo-code of RRA is shown in Algorithm 1. First, the set $M$ is sorted in descending order with respect to the number of antennas required by each UE. After that, the resource allocation is performed for each UE in turn (line 2). In the case that UEs with close distance are not interfered with by each other, the available RB set $RB_m$ should be determined to satisfy the constraint of Equation (4) (line 3). Then, under the constraint of Equations (2), (3), (5), (9), and (20), a RU set $J$ in $R$ and a corresponding antenna set $K$ in $A$ are found, where the antennas in $K$ have the same $rb_m$ in $RB_m$. The eligible sets $J$ and $K$ are the RU set and antenna set allocated for

UE $m$, respectively (lines 4–5). If there are RUs in $J$ that are not assigned wavelengths, a wavelength $l$ will be randomly selected from $L$ and be judged whether it has enough bandwidth to accommodate the bandwidth requirement of UE $m$. If yes, wavelength $l$ will be allocated to these RUs; otherwise, $l$ is removed from $L$, and a wavelength is randomly selected from $L$ to repeat the above steps (lines 6–14). Once UE $m$ is allocated, the available bandwidth $B_l$ for each wavelength in $L$ is updated (line 15). The algorithm ends after all the UEs in $M$ have been allocated.

The pseudo-code of RRA algorithm.

---

**Algorithm 1:** Random Resource Allocation Algorithm (**RRA**)

---

**Input:** UE set $M$, RU set $R_m$ for UE $m$, antenna set $A$, RB set $RB$, wavelength set $L$, available bandwidth $B_l$ for wavelength $l$
**Output:** allocated RU set $G_m^R$, allocated antenna set $G_m^A$, allocated RB set $G_m^{RB}$, allocated wavelength set $G_r^L$
**Initialize** : $G_m^R \leftarrow \varphi$ ; $G_m^A \leftarrow \varphi$ ; $G_m^{RB} \leftarrow \varphi$ ; $G_r^L \leftarrow \varphi$

| | |
|---|---|
| 1. | Sort $M$ in random order of $k_m$. |
| 2. | **for** $m \in M$ **do** |
| 3. | Find the available RB set $RB_m$ for $m$ that satisfies Equation (4). |
| 4. | Find a RU set $J$ in $R_m$ and a corresponding antenna set $K$ in $A$, where the antennas in $K$ have the same $rb_m$ in $RB_m$ that satisfies Equations (2), (3), (5), (9), and (20). |
| 5. | $G_m^R \leftarrow J$, $G_m^A \leftarrow K$, $G_m^{RB} \leftarrow rb_m$. |
| 6. | **if** there are RUs in $J$ that are not assigned wavelengths, **then** |
| 7. | Select a wavelength $l$ in $L$ randomly. |
| 8. | **while** $B_l + r_m \leq R_o$ **do** |
| 9. | $G_r^L \leftarrow l$. |
| 10. | **else** |
| 11. | $L \leftarrow L - \{l\}$. |
| 12. | Select a wavelength $l$ in $L$ randomly. |
| 13. | **end while** |
| 14. | **end if** |
| 15. | Update $B_l$ of each wavelength in $L$. |
| 16. | **end for** |
| 17. | Allocate $G_m^R$, $G_m^A$, and $G_m^{RB}$ to UEs and allocate $G_r^L$ to RUs. |

---

The randomness of the RRA algorithm may cause more wavelengths to be occupied and more fronthaul bandwidth to be wasted. The Fronthaul Occupancy Minimized Resource Allocation (FOM-RA) Algorithm 2 is proposed to tackle this issue. The pseudo-code of the FOM-RA algorithm is shown in Algorithm 2. The certain steps of the procedure, i.e., the sorting of UE and the determination of $RB_m$ (lines 1–3), are the same as in RRA. For UE $m$ in $M$, if a RU set $J$ can be found in $R_m\prime$ and a corresponding antenna set $K$ can be found in $A$ under the constraint of Equations (2), (3), (5), (9), and (20), where the antennas in $K$ have the same $rb_m$ in $RB_m$, $J$ and $K$ are the RU set and antenna set allocated for UE $m$, respectively (lines 4–5). If no sets are found, $J$ and $K$ will be selected from $\{R_m - R_m\prime\}$ and $A$, respectively. Then the available $rb_m$ are selected from $RB_m$. After that, the set $R_m\prime$ is updated (lines 7–9). In the wavelength allocation stage, if the occupied bandwidth of the first wavelength $\lambda$ in wavelength set $L$ is less than half of $R_o$, wavelength $\lambda$ will be allocated to these RUs, which is denoted as $G_r^L$; otherwise, the next wavelength $(\lambda + 1)$ in $L$ will be allocated and $\lambda$ will be removed from $L$ (lines 10–15). Once UE $m$ is allocated, the available bandwidth $B_l$ for each wavelength in $L$ will be updated (line 17). The algorithm ends after all the UEs in $M$ are allocated.

The pseudo-code of FOM-RA algorithm.

---

**Algorithm 2:** Fronthaul Occupancy Minimized Resource Allocation Algorithm (**FOM-RA**)

---

**Input:** UE set $M$, RU set $R_m$ for UE $m$, antenna set $A$, RB set $RB$, wavelength set $L$, available bandwidth $B_l$ for wavelength $l$
**Output:** Occupied RU set $R_m\prime$, allocated RU set $G_m^R$, allocated antenna set $G_m^A$, allocated RB set $G_m^{RB}$, allocated wavelength set $G_r^L$
**Initialize**: $G_m^R \leftarrow \varphi$; $G_m^A \leftarrow \varphi$; $G_m^{RB} \leftarrow \varphi$; $G_r^L \leftarrow \varphi$; $R_m\prime \leftarrow \varphi$

| | |
|---|---|
| 1. | Sort $M$ in descending order of $k_m$. |
| 2. | **for** $m \,\epsilon\, M$ **do** |
| 3. | Find the available RB set $RB_m$ for $m$ that satisfies Equation (4). |
| 4. | **if** find a RU set $J$ in $R_m\prime$ and a corresponding antenna set $K$ in $A$, where the antennas in $K$ have the same $rb_m$ in $RB_m$ that satisfies Equations (2),(3),(5),(9) and (20), **then** |
| 5. | $G_m^R \leftarrow J$, $G_m^A \leftarrow K$, $G_m^{RB} \leftarrow rb_m$. |
| 6. | **else** |
| 7. | Find a RU set $J$ in $\{R_m - R_m\prime\}$, a corresponding antenna set $K$ in $A$, and an available RB set $rb_m$ in $RB_m$ that satisfies (2),(3),(5),(9) and (20). |
| 8. | $G_m^R \leftarrow J$, $G_m^A \leftarrow K$, $G_m^{RB} \leftarrow rb_m$. |
| 9. | $R_m\prime \leftarrow R_m\prime + \{J\}$. |
| 10. | **if** $B_\lambda < R_o \cdot 0.5$, **then** |
| 11. | $G_r^L \leftarrow \lambda$. |
| 12. | **else** |
| 13. | $G_r^L \leftarrow \lambda + 1$. |
| 14. | $L \leftarrow L - \{\lambda\}$. |
| 15. | **end if** |
| 16. | **end if** |
| 17. | Update $B_l$ of each wavelength in $L$. |
| 18. | **end for** |
| 19. | Allocate $G_m^R$, $G_m^A$, and $G_m^{RB}$ to UEs and allocate $G_r^L$ to RUs. |

---

## 5. Simulation Results and Analysis

In this section, we present numerical results based on the simulations to evaluate the performance of the ILP and heuristic algorithms. The CPLEX linear programming solver in AMPL and MATLAB tools are used to solve our ILP mathematical model and heuristic algorithms, respectively. In our simulations, two scenarios are considered: the small-scale network and the large-scale network.

### 5.1. Simulation Parameter Setup

In our simulations, we consider a TWDM-PON-enabled CF-mMIMO system with 1 OLT and 375 ONUs, and each ONU is connected with one RU. As for the setup of the wireless part, the RB has a frequency range of 180 kHz, and each of them has 12 subcarriers and 14 OFDM symbols within a transmission time interval (TTI). The modulation and coding scheme (MCS) level is 7.

The simulation parameters for the two scenarios are as follows. In the scenario of the small-scale network, four wavelengths are contained in the TWDM-PON, and each of them has a capacity of 0.5 Gb/s. In each RU, the number of antennas is two, and each antenna has 16 RBs. The number of UEs ranges from 10 to 50. For each UE, the number of requested RBs is randomly selected in the range of 3 to 5, and the number of requested antennas is randomly selected in the range of 8 to 12. In the scenario of the large-scale network, 16 wavelengths are contained in the TWDM-PON, each of which has a capacity of 15 Gb/s. The number of antennas in each RU is four, and each antenna in RUs has 128 RBs. The number of UEs ranges from 90 to 150. For each UE, the number of requested RBs is randomly selected in the range of 24 to 40, and the number of requested antennas is randomly selected in the range of 16 to 24. Additionally, our simulations are run by a computer with a CPU of Intel Core i5-12400X and a memory of 256 GB. In the simulations, 10 sets of demand are tested in the ILP and heuristics algorithms, and the results are averaged over these 10 sets.

### 5.2. Scenario 1: The Small-Scale Network

To investigate the network scalability, we mainly focus on the fronthaul bandwidth occupancy and the number of occupied wavelengths in the small-scale network. Simulation results are shown in Figures 4 and 5.

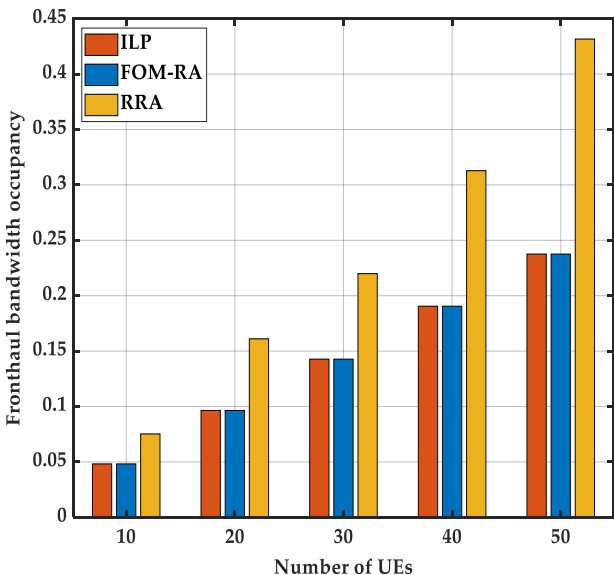

**Figure 4.** Fronthaul bandwidth occupancy versus the number of UEs in the small-scale network.

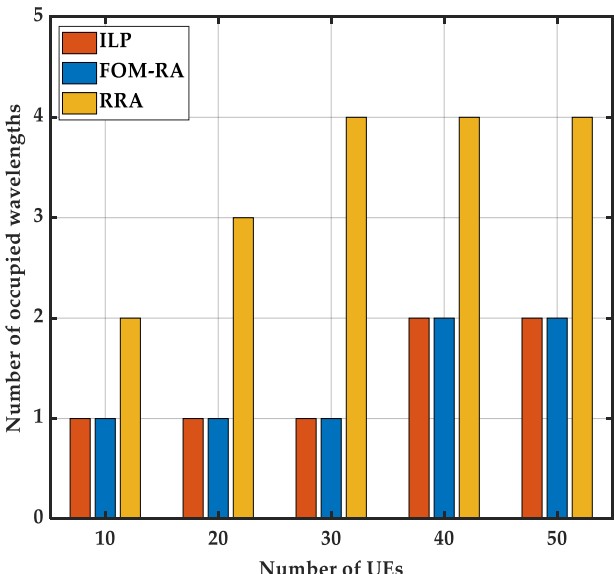

**Figure 5.** The number of occupied wavelengths versus the number of UEs in the small-scale network.

Figure 4 exhibits the fronthaul bandwidth occupancy versus the number of UEs in the small-scale network. Remarkably, the proposed FOM-RA algorithm attains equivalent outcomes to those of the ILP approach. In comparison with RRA, both FOM-RA and ILP occupy less bandwidth. When the number of UEs rises up to 50, the bandwidth occupancy in FOM-RA is reduced by 73% compared with RRA.

The occupied wavelengths of the two algorithms exhibit similar performance compared to the required bandwidth, as shown in Figure 5. It is evident that our proposed RRA algorithm achieves the same results as ILP. As the number of UEs rises, only one or two wavelengths are occupied in both ILP and FOM-RA algorithms. This outcome stems from the low demand for the fronthaul bandwidth, whereby a limited number of wave-

lengths can effectively cater to all requirements. As for RRA, compared with the FOM-RA algorithm, it occupies significantly more wavelengths due to its random selection policy.

Additionally, the execution time required by the algorithms is also taken into account. Table 1 shows the execution time of heuristic algorithms in the small-scale network. It can be seen that, even within the context of a small-scale network, ILP demonstrates considerably longer execution time compared to the heuristic algorithms. Therefore, it is inefficient to solve the large-scale network with ILP. The executing time of two heuristic algorithms is within 0.5 s.

**Table 1.** The executing time of ILP and heuristic algorithm in the small-scale network.

| Number of UEs | Executing Time of ILP | Executing Time of FOM-RA | Executing Time of RRA |
|---|---|---|---|
| 10 | 40 s | 0.21 s | 0.16 s |
| 20 | 15 min | 0.23 s | 0.18 s |
| 30 | 2 h | 0.19 s | 0.24 s |
| 40 | 24 h 30 min | 0.23 s | 0.31 s |
| 50 | 48 h 10 min | 0.18 s | 0.33 s |

*5.3. Scenario 2: The Large-Scale Network*

In the scenario of the large-scale network, not only the number of wavelengths and their capacities are extended in the TWDM-PON, but the number of UEs increases significantly as well, which imposes higher requirements on the network scalability. Similar to Scenario 1, we are focusing on two aspects: the required fronthaul bandwidth and the number of occupied wavelengths. Due to the long execution time of ILP, only the performances of two heuristic algorithms are investigated.

Figure 6 shows the fronthaul bandwidth occupancy versus the number of UEs in the large-scale network. It can be seen that a similar trend for two algorithms is obtained compared to the small-scale network, where the proposed FOM-RA algorithm can effectively reduce the fronthaul bandwidth occupancy compared to RRA. This can be attributed to FOM-RA's strategic preference for selecting occupied RUs and antennas, whereas RRA's selection process is random and encompasses all available RUs and antennas. In addition, the fronthaul bandwidth occupancy is growing as the number of UEs increases.

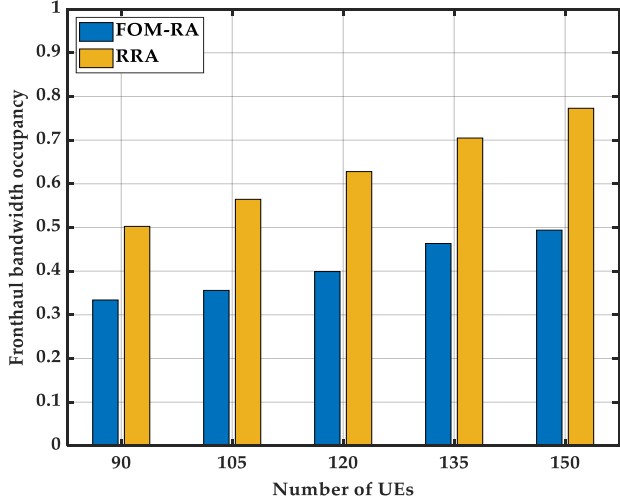

**Figure 6.** Fronthaul bandwidth occupancy versus the number of UEs in the large-scale network.

Figure 7 shows the number of occupied wavelengths versus the number of UEs in the large-scale network. Compared with RRA, FOM-RA occupies fewer wavelengths. When the number of UEs is large, RRA occupies all 16 wavelengths in TWDM-PON. This can be attributed to FOM-RA's strategic preference for selecting occupied RUs and

antennas, whereas RRA's selection process is random and encompasses all available RUs and antennas.

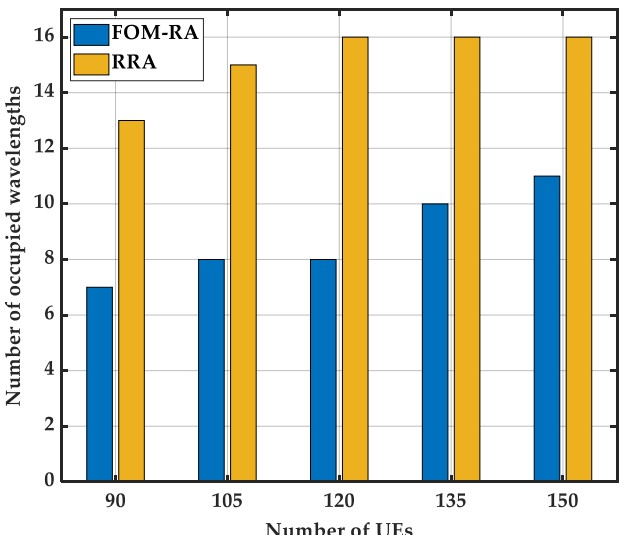

**Figure 7.** The number of occupied wavelengths versus the number of UEs in the large-scale network.

Specifically, the radio RB utilization in the large-scale network is also investigated to exhibit the performance of wireless resource allocation, which is defined as the ratio of the number of occupied RBs to the total number of RBs in the occupied antennas. Figure 8 shows that the radio RB utilization in FOM-RA stably remains higher than that in RRA, regardless of the number of UEs. This is because FOM-RA occupies a reduced number of RUs, consequently leading to a higher utilization of RBs within each RU to cater to UEs.

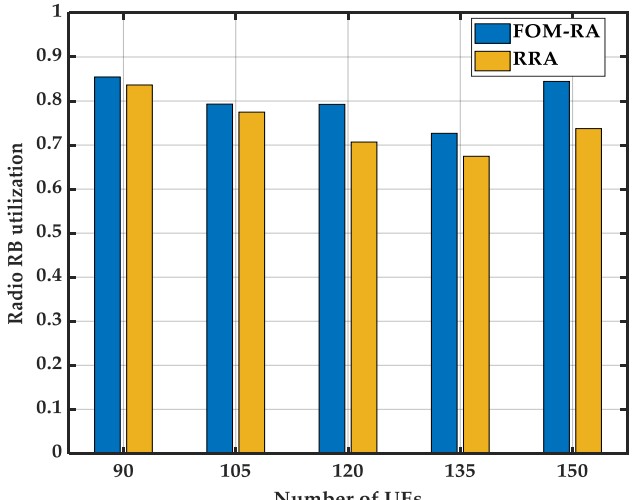

**Figure 8.** The radio RB utilization versus the number of UEs in the large-scale network.

Table 2 shows the execution time of two heuristic algorithms in the large-scale network. It can be seen that the executing time in both algorithms is within 2 s. In addition, the executing time of FOM-RA is less than that of RRA because the complexity of RRA is higher than that of FOM-RA in the large-scale network.

**Table 2.** The executing time of two heuristic algorithms in the large-scale network.

| Number of UEs | Executing Time of FOM-JRA | Executing Time of RRA |
|:---:|:---:|:---:|
| 90 | 0.95 s | 1.32 s |
| 105 | 1.19 s | 1.38 s |
| 120 | 1.25 s | 1.68 s |
| 135 | 1.16 s | 1.65 s |
| 150 | 1.12 s | 1.99 s |

## 6. Conclusions

In this paper, a TWDM-PON-enabled CF-mMIMO system is considered. In this system, the allocation of both optical and wireless resources (including wavelength, antenna, RU, and radio RB) plays an important role in the system's scalability. Based on this, we introduce the system architecture in detail and present a joint resource scheduling method in this system. Furthermore, an ILP model is formulated, and two heuristic algorithms (i.e., RRA and FOM-RA) are proposed, which aim at minimizing the fronthaul resource occupancy, thereby increasing the network scalability. We simulated ILP and two heuristics under small-scale and large-scale network scenarios, and some results are discussed. Simulation results show that both algorithms can significantly reduce the execution time in small-scale and large-scale networks. Notably, our proposed FOM-RA algorithm can achieve the same performance as the ILP and outperform the RRA algorithm in terms of fronthaul resource occupancy and radio RB utilization. The joint upstream resource scheduling in the TWDM-PON-enabled CF-mMIMO system will be further investigated in future works.

**Author Contributions:** Conceptualization, T.X., K.A.M. and C.L.; methodology, T.X. and K.A.M.; software, T.X.; validation, T.X., K.A.M. and C.L.; formal analysis, T.X.; investigation, T.X. and K.A.M.; resources, T.X.; data curation, T.X.; writing—original draft preparation, T.X.; writing—review and editing, K.A.M. and C.L.; visualization, C.L.; supervision, C.L.; project administration, C.L.; funding acquisition, C.L. All authors have read and agreed to the published version of the manuscript.

**Funding:** This work was supported in part by the National Key Research and Development Program of China (2020YFB1807201), the National Natural Science Foundation of China (62171119), and the Key Re-search and Development Plan of Jiangsu Province (BE2021013-3).

**Institutional Review Board Statement:** Not applicable.

**Informed Consent Statement:** Not applicable.

**Data Availability Statement:** Data is unavailable due to privacy.

**Conflicts of Interest:** The authors declare no conflict of interest.

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
