# Peer review of "Joint Resource Allocation in TWDM-PON-Enabled Cell-Free mMIMO System"

_photonics, doi:10.3390/photonics10111180_

Round 1

Reviewer 1 Report

In general, the novelty and the quality of this paper is good.  The author proposed a joint scheduling method of wavelength, antenna, radio unit (RU), and radio resource block (RB) resources in the TWDM-PON enabled CF-mMIMO system.  Furthermore, an integer linear programming (ILP) model for joint resource allocation is proposed to minimize the fronthaul resource occupancy, thereby increasing network scalability.  However, there are still several small issues to be addressed before it can be accepted.
1.     Grammatical errors should be avoided
2.     The places of figures need to be adjusted to make it easy reading.
3.     It is better to introduce future work in conclusion.

Reviewer 2 Report

160: How do the results you achieved depend on the complexity of the services on individual UEs. Will the Bit Rate requirements on the UEs affect your calculated results?

304: What was the reason for the simulation for the small-scale network with only 4 wavelengths and each of them has a capacity of 0.5 Gb/s only? 0.5 Gb/s is not the typical capacity for PONs in general. 

I recommend 

325: For Fig 4 write:  Fronthaul bandwidth occupancy versus the number of UEs in the small-scale network.

329: For Fig 5 write:  The number of occupied wavelengths versus the number of UEs in the small-scale network.

... similar for Fig.6 and Fig 7

366: Executing time is dependent on many parameters - give info about the computer power you used for calculations

Reviewer 3 Report

The authors have proposed a joint resource allocation scheme in a TWDM-PON enabled CF-mMIMO system to increase the network scalability. The research is worth publishing however it can be increase the quality of paper by addressing following lacks.

1. In section 3.3, the variables and defined however, the equations are not elaborated. It is highly suggested that explain each and every equation so the readers can understand your concept.

2. What is Objective function? Why it is needed?

3. Elaborate Heuristic Algorithm in section 4. Why these two techniques are used? What are the benefits and how it is better than previous techniques. 

4. Also the literature review is very less. Only 20 references.. I also notice that key papers are not cited. I highly recommend to read the literature and add more citations of key papers of MIMO. I advice to add following paper and also other papers showing that your research is better than the published work.

Holographic MIMO surfaces for 6G wireless networks: Opportunities, challenges, and trends

C Huang, S Hu, GC Alexandropoulos… - IEEE Wireless …, 2020 - ieeexplore.ieee.org  

An Overview of Massive MIMO for 5G and 6G

FAP de Figueiredo - IEEE Latin America Transactions, 2022 - ieeexplore.ieee.org   Shallow water acoustic channel modeling and MIMO-OFDM simulations G Qiao, Z Babar, L Ma, L Wan, X Qing, X Li, M Bilal 2018 15th international bhurban conference on applied sciences and …

The quality of English is up to mark however it can be further polished.
